# Assessment of Ecological and Economic Efficiency of Agroforestry Systems in Arid Conditions of the Lower Volga

## Evgenia A. Korneeva * and Alexander I. Belyaev

Federal Scientific Center of Agroecology, Complex Melioration and Protective Afforestation, Russian Academy of Sciences, University Ave, 97, 400062 Volgograd, Russia
\* Correspondence: korneeva.eva@list.ru; Tel.: +7-9178407904

**Abstract:** The aim of this study was to research the cost effectiveness of creating forest reclamation complexes on slopes, as well as to determine the patterns of their orographic dynamics, taking into account environmental aspects in arid conditions. With the help of modeling agroforestry landscapes, we established forest plantations created from Lanceolate ash (*Fraxinus lanceolata*) in arid climatic conditions on sloping lands, the cost of planting of which is EUR 1202–EUR 1453 per ha of forest. The specific capital intensity of the arrangement of land use by forest stands is EUR 24–EUR 63 per hectare of afforested plot, while 5–11% accounts for the cost of logging of forest care and 2–30% for the inclusion of a hydraulic element in forest reclamation systems. The monetary equivalent of the return on these investments in the form of prevented damage from soil erosion and air pollution is EUR 333–EUR 940 per hectare of afforested plot per year. This economic effect increases with the growth of the protective forest cover of the plot (by reducing the interband space) by almost 3 times. The benefit–cost ratio for all forest reclamation strategies on slopes is greater than 1, which confirms the high efficiency and expediency of capital investments in forest reclamation activities on slope lands to preserve the land resources of various regions.

**Keywords:** agroforestry; slope lands; bioengineering parameters; protective forest cover; forest reclamation strategy; costs; ecosystem services; benefit-cost ratio

## 1. Introduction

The concern of the world scientific community about the conclusions of climate change and the problems of sustainable land use have put forward new requirements for agriculture. The world's leading economies have shifted their attention from maximizing production to environmentally friendly agriculture. The integration of forests into agricultural systems—agroforestry—is recognized as a key advantage and the main option for sustainable land resources management in agricultural regions. A number of leading countries have reoriented their forest policy to include this holistic system of sustainable land use [1].

Afforestation of arable land is not something new, but it is one of the very important initiatives to improve land resources management in the world. Today, almost a billion hectares of agricultural lands have a protective forest cover of more than 10 percent, and according to available estimates [2], a total of 1.6 billion hectares of land will be protected by trees in the foreseeable future.

Agroforestry is defined as the deliberate integration of woody vegetation (trees and shrubs) with agricultural crops and/or livestock on a land management unit of any scale (for example, a plot, farm, landscape, etc.) [3]. Agroforestry is a symbiosis of growing trees, crops and livestock, where each component benefits each other [4]. Sustainable reforested land use means providing by forests with benefits for land users in the form of ecosystem goods and services in the long-term perspective [5].

It is noted [6] that as soon as a farmer implements agroforestry, they can see an improvement in the condition of the soil and other ecosystem functions, such as improved

water infiltration and reduced nutrient runoff, which then increases crop yields or reduces production costs and, consequently, increases returns.

Agroforestry can also prevent environmental degradation, increase agricultural productivity, increase carbon uptake, and maintain healthy soil and healthy ecosystems while providing stable income and other benefits for human well-being [7].

Recent special studies [8] show that agroforestry systems have significant potential to achieve numerous sustainable development goals. At the same time, environmental and social sustainability goals are being achieved especially successfully by improving land use efficiency, expanding employment opportunities on farms and interacting with local communities. At the same time, it is noted that there are still significant financial barriers that may hinder the further implementation of agroforestry.

In the era of green business and environmental sustainability, Russia cannot ignore global trends in science and production, because it has an undesirable state of land resources similar to most countries. However, the problem of the creation of protective forest stands has been present and almost unsolved throughout its modern history [9]. The introduction of private ownership of agricultural land logically led to the full responsibility of the owners for their condition. At the same time, in a competitive economy, the additional costs of maintaining protective forest plantations began to be considered by them as an additional burden. The situation could not be corrected by scientific assessments based on determining the raising in crop yields during afforestation of fields [10].

Indeed, in modern Russian economic conditions, the costs of creating plantations are not covered by an increase in yield, especially due to inflationary processes in the country. Therefore, this article proposes to develop new elements of motivation related to the monetary assessment of ecosystem services of forest plantations that ensure the preservation of soil resources and clean air and are still considered free. This will have an effect, since sustainability issues are currently more important for Russian society than crop margins.

The article is based on the transformation of the approach to estimating the economic efficiency of forest reclamation, when it is considered primarily as a unique resource-saving component of agricultural production and only lastly as an element of land use that provides crop growth. Traditional methods of assessing the effectiveness of protective forest plantations—simple descriptive accounting systems for additional agricultural products obtained in forested fields—are supplemented with new analysis tools adopted in the world practice of environmental economics [11–13] and adequate for Russian regional realities [14]. With the help of these tools, it became possible to expand the boundaries of the concept of "agroforestry efficiency" and to determine the main bioengineering parameters under which it becomes an acceptable option for the land user; that is, it is an environmentally and economically beneficial measure that promotes the growth of efficiency of agricultural production.

The main purpose of this study was to research the cost effectiveness of creating forest reclamation complexes on slopes, as well as to determine the laws of their orographic dynamics, taking into account environmental aspects.

## 2. Materials and Methods

### 2.1. Case Study Sites

The research was conducted in relation to the dry steppe zone located within the Volgograd region of Russia (Figure 1). Geomorphologically, the zone is represented by the southeastern end of the Volga Upland and the Ergeny Upland. The predominant soils here are chestnut soils. According to their mechanical composition, they are clayey and heavy loamy with a content of silty fraction from 23 to 41% [15].

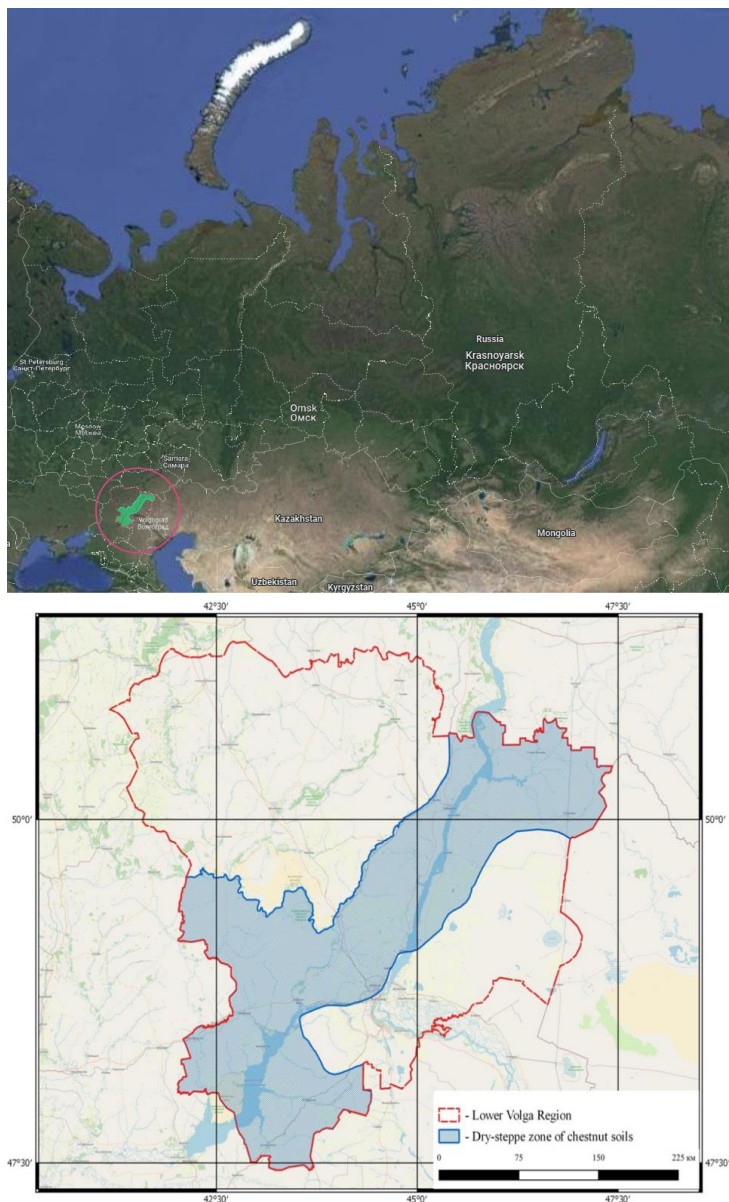

**Figure 1.** Geographical location of the dry-steppe zone of chestnut soils.

Humus in typical chestnut soils noticeably decreases from the northeast to the southwest of the dry-steppe zone; therefore, it is most often observed in chestnut soils of the Volga upland (2.4%) and least of all in soils of the Ergeny upland (2.2%). The content of gross forms of nitrogen (N) is 0.2% and 0.15%, respectively; phosphorus ($P_2O_5$)–0.05% and 0.09%; potassium ($K_2O$)–1.7% for all geomorphological areas [15].

The climate of the dry-steppe zone is very continental (the coefficient of continence is 209–224). This is manifested in moderately cold and snowless winters, as well as warm and very dry summers. The sum of active temperatures is 2500–3100 °C. The peculiarity of the zone is the discrepancy between the amount of annual precipitation and possible evaporation. Thus, the annual precipitation is 255–443 mm, and the evaporation rate is about 2–3 times greater. According to the annual moisture content, the dry-steppe zone belongs to a very arid and semi-dry zone. The moisture coefficient is 0.26–0.48 [16].

The 5-year dynamics (2016–2020) of the main climatic indicators, obtained on the basis of data from reference weather stations of the dry-steppe zone, indicates a high continentality of the climate of the studied region, increasing from the north-east to the south-west (Figure 2).

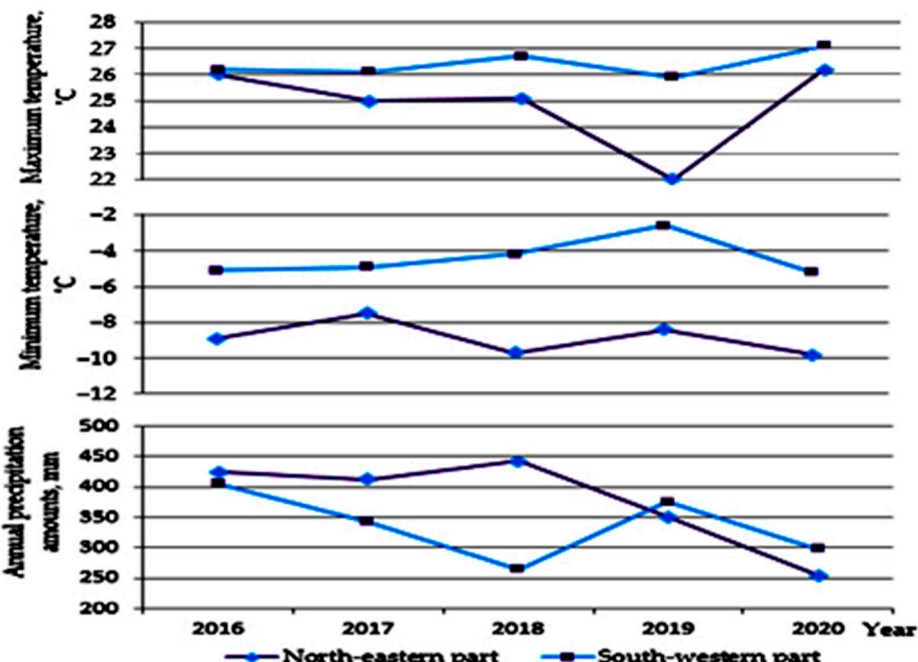

**Figure 2.** Climatic characteristics of the dry steppe zone of the Lower Volga region (according to weather stations in the cities of Kamyshin and Kotelnikovo). Location of the meteorological station in Kamyshin (north-eastern part): latitude 50.07, longitude 45.37. Location of the meteorological station in Kotelnikovo (south-western part): latitude 47.63, longitude 43.15. Adapted with permission from [17].

In the dry-steppe zone, forests, mainly floodplain forests, occupy only 3.4%. For the most part of the zone's territory is used for agricultural land. The main branches of agriculture are the cultivation of cereals (winter wheat, corn, etc.), vegetables, oilseeds and melons [18].

The index of total land degradation in the Volgograd region is 52 points. The predominant degradation process in the dry-steppe zone is water erosion of soils, which is related to the presence within its limits of steep hills with pronounced relief—the leading factor of water erosion [19].

Technological Parameters of the Placement of Protective Forest Stands within the Agricultural Landscape

The dry-steppe conditions of the Lower Volga region mean it is a territory where there is a high risk of the formation of water erosion. On the one hand, this process is facilitated by the conditions of arid climate, which determine the nature of rains, when the most intense precipitation falls before the beginning of the vegetation of agricultural plants and causes significant damage to the soil. On the other hand, plowed slopes with light soils and a slope of more than 3°, even with a low intensity of rains, also remain indented streambeds.

The fight against the processes of water erosion in the region is done using of highly effective agrotechnical techniques (contour tillage, mulching, strip farming); however, these measures are not enough for a sustainable long-term farming system. The existing practice of soil protection from erosion [20] indicates that land use in such terms is often unprofitable and requires a higher level of protection of problem lands—the creation of flow-regulating forest strips, and on steep slopes also, the construction of the simplest hydraulic structures.

Based on the current standards for the creation of protective forest plantations on the slopes [21,22], optimal bioengineering and technological parameters of the forest-reclamation arrangement of erosion-hazardous arable lands have been developed in relation

to the regional conditions of the dry-steppe of the Volgograd region (Table 1). It has been established that the most typical size of an agricultural farm in this zone is 1000 m × 1000 m.

**Table 1.** Bioengineering and technological models of reclaimed forest land object on sloping lands in the dry steppe zone using Lanceolate ash (100 ha).

| Calculated Indicators | Open Slope | Slope Steepness, ° | | |
| --- | --- | --- | --- | --- |
| | | 2.1–3.0 | 3.1–5.0 | 5.1–6.0 |
| Forest stand—forest stand space, meters | x | 520 | 320 | 190 |
| Width of forest stands, meters | x | 9 | 9 | 6 |
| Plot area, ha: | | | | |
| without forest plantations | 100 | x | x | x |
| occupied by forest plantations | x | 1.73 | 2.81 | 3.16 |
| under agricultural land | 100 | 98.27 | 97.19 | 96.84 |
| The simplest hydraulic structures (in the lower aisle of the forest plantation) | x | the collapse of the forest strip | shaft–ditch height of 0.5–0.7 m | shaft–ditch height of 0.9–1.1 m |
| Protective forest cover, % | x | 1.76 | 2.89 | 3.26 |
| Crop rotation in a forested area | x | grain and steam | | grass and steam |

For satisfactory erosion protection of this site, the system Silvo-arable Agroforestry is proposed—with extended protection from live trees planted and maintained to protect agricultural land from harmful runoff. This system consists in creating narrow forest strips of solid rocks across the slope, which have an advantage over the use of precocious rocks (due to a longer service life [5]) and do not require soil fertility.

So, in dry-steppe areas on washed chestnut soils, Lanceolate ash is recommended for planting. The planting scheme for this breed is as follows: Ash L.–Ash L.–Ash L. Forest plantations are created from three rows. Seating capacity–3 m × 1.5 m. The width of the forest stands is 9 m. The density of trees per hectare is 2222 pieces.

The addition of forest strips with the simplest hydraulic structures (by placing them in the lower aisle and increasing the height of structures with increasing the steepness of the slope) allows the reduction of the number of forest strips from 3 to 2 rows and brings protection from water erosion to 100% [20,23].

Forest plantations in arid conditions are created by planting 1-year-old seedlings on an annual black pair with a plowing depth of 50–60 cm. Soil care is carried out by forest and agricultural cultivators. In the first 2 years, the soil is loosened 4–6 times, in the 3rd year 3–5 times, and in subsequent years 1–2 times before the crowns are closed [24].

Lanceolate ash grows up to 12 m high, is not very demanding to the soil, and can grow on sandy loams. It is characterized by great drought resistance, as a result of which it is recommended for the afforestation of plains and slopes. It grows quickly, even in arid areas. The standard service life for the climatic conditions of the dry steppe for Lanceolate ash is 35 years.

It is extremely impractical to use another long–lasting breed, the Petiolate oak (*Quercus robur*), for the conditions of sloping lands, since it is a very demanding breed for soil quality and there is a risk of early death of trees on washed-out soils [25].

### 2.2. Modeling of Agroforestry Systems and Land Use

The effectiveness of agroforestry was studied on the constructed technological models of forest-reclaimed land use with specified bioengineering parameters that provide satisfactory protection of the soil from natural anomalies [21]. The spatial influence of protective forest stands on the adjacent territory was studied using a system analysis [26]. The area of forest plantations and the territory of the protected landscape were modeled. Ultimately, the protective forest cover of the site was determined—the ratio of the area of forest plantings to the total land use area.

Regional models of agroforestry complexes included various combinations of systems of flow-regulating protective forest plantations and land use located on slopes. The main bioengineering parameters for these models (the range of forest-forming species, the number on the site, the width and row of protective forest stands, inter-lane spaces) were established on the basis of basic recommendations for afforestation of sloping agricultural lands [27–29].

The constructed engineering and technological models of runoff-regulating forest strips imitated a variety of ratios "convex profile of the catchment area—forest—land use". Using these models, the efficiency of forest reclamation was analyzed for various options for placing forest stands on slopes with steepness from 2.1 to 6.0 degrees.

The "Runoff-regulating" forest reclamation strategy on slopes from 2.1 to 5.0 includes a short-rotation grain–steam crop rotation on interband fields with the following structure: black steam (25%)–grain crops (75%). On slopes of more than 5.1, the forest reclamation strategy provides for the following crop rotation between trees: Sudanese grass—perennial grasses of the 1st year of use—perennial grasses of the 2nd year of use—perennial grasses of the 3rd year of use—winter rye for green fodder [30].

*2.3. Data Collection*

The main research method is the construction of simulation models of crop rotation fields with different protection of forest strips and a systematic analysis of the data obtained. Fields equipped with forest strips were simulated, both in accordance with the requirements of current regulations [21,22] and taking into account the latest achievements of science [27]. Based on secondary data on the level of fertility in protected cells and the amount of dust filtered by forest plantations in the agroforestry system, key performance indicators of the "Runoff-regulating" forest reclamation strategies were calculated.

Information about the potential anti-erosion effect at different distances of trees from each other on sloping land plots was obtained as a result of reliable data from the literature, as well as long-term field studies of the Erosion Department and the Economics Department of the Institute of Agroforestry—VNIALMI (Volgograd, Russia).

When calculating the anti-erosion effect, we proceeded from the fact that in cells protected by forest strips, damage from soil flushing is prevented to the limits known to science [23]. So, the steeper the slope, the greater the mass of organic materials and nutrients are prevented as a result of flushing and the greater the effect. Thus, the reference value of organic matter (on average 2.3%) and the ethylene gross forms of NPK nutrients (0.2 (N), 0.09 (P) and 1.7% (K), respectively) in chestnut non-eroded soils in areas protected by forest plantations were taken as a basis. Further, a decrease in these indicators was estimated during erosion in open fields under the condition [31]: the decrease in the thickness of the soil profile (A + B) with a weak degree of erosion is 3–25%, with an average degree of 26–50%, with severe erosion being 51–75%. The decrease in humus and nutrient reserves in the soil profile (A + B) decreases by 11–20% with a low intensity of harmful runoff, by 21–40% with an average intensity and by 41–80% with a strong intensity of erosive runoff.

The indicated percentage decrease in humus and nutrients was further converted into tons for chestnut soils and the mass of these losses per hectare was obtained. Then, using the compensation cost method, they were evaluated in monetary terms.

When calculating the effect of dust accumulation by protective forest plantations, secondary data [32] were used, in obtaining which the method of aspiration of dusty air was used, followed by weighing of deposited dust particles. It was found that the average amount of dust deposited by $m^2$ of Lanceolate ash foliage in the agroforestry system is 2.55 g. The amount of dust deposited by the foliage of 1 tree in the period between rains is 497.3 g. Multiplying this value by the number of trees on the site and the number of periods between rains per year, biophysical data on the amount of dust filtered by a hectare of forest stands were obtained.

Monetary assessment of the air-cleaning function of agroforestry systems was made using the methodology [33] at the rate of payment for emissions [34].

*2.4. Data Analyses*

2.4.1. Tools for Economic Analysis of Agroforestry Efficiency

The analysis of the ecological and economic efficiency of agroforestry was carried out in several stages with a set of the following assessment tools:

Stage 1. Analysis of regional environmental risks.

Identification of environmental risks in the justification of forest reclamation measures is the starting point of all studies. It is known that a specific regional forest management strategy is being developed for a specific territory, which makes it possible to neutralize the prevailing natural anomalies here as much as possible. So, in the territory with a high level of danger of wind erosion, it is recommended to create wind-breaking forest strips, water erosion runoff-regulating forest strips. These forest belts differ significantly in terms of the conditions for the creation of forest crops, the productivity and durability of full-fledged plantings, and, ultimately, the entire effectiveness of forest reclamation [35]. This determines their optimal placement in the fields (range of breeds, lane spacing), as well as bioengineering parameters (length, width, row, pitch, and planting method).

Stage 2. Budgeting of regional costs.

Cost budgeting includes an assessment of the projected costs for the entire range of afforestation works on land plots. Thus, these costs include the costs of creating forest plantations, their cultivation, maintaining forest plantations in the necessary functional condition and their operation. The cost of these works was determined using the current calculation and technological maps for landscaping and the creation of protective forest plantations [24] with a transition to prices in 2022 on the basis of special indices of changes in the cost of construction and installation works in forestry production, developed by the Construction Ministry of the Russian Federation (Moscow, Russia).

Stage 3. Analysis of the economic cost of possible environmental effects (ecosystem services) of agroforestry.

When assessing the positive impact of protective plantations on the agricultural territory, it was found that there is no unified classification and generally accepted mechanism for assessing ecosystem services of agroforestry in Russia. Each land user decides for themselves which tree utilities to use for their target program. We start from the fact that a narrow forest strip can have the most significant impact on the land use area in the form of the influence of trees on the preservation of soil fertility and environmental cleanliness. Thus, we have limited ourselves to studying the regulatory ecosystem services of trees—this is the control of erosion and atmospheric pollution in the specific context of agricultural systems.

The cost equivalent of these ecosystem services was estimated using the known methods [36–39] at prices of 2022. So, to assess the cost of regulating ecosystem services (regulation of erosion and air quality), replacement cost methods are usually used when they go from the opposite and consider how much a person will have to invest in the restoration of the territory and air purification if the ecosystem is disrupted and stops providing self-regulation services. When using this method, the main assumption is that the cost of replacing an ecosystem good or service with such an alternative or substitute can be taken as an indicator of its value in terms of saved costs. In this study, the values of soil and air quality in agroforestry systems were assessed by analyzing the costs of replacing these services with artificial means.

This equivalent was determined in RUB and converted into EUR at the exchange rate set by the Central Bank, Moscow, Russia, on 10 April 2022.

Stage 4. Time factor analysis.

The economic effect of trees varies from a negative value (the year of allocation of the area for forest plantations) to a positive value, increasing in subsequent years of growth of forest crops. This determines the use of the time factor and the growth dynamics of the main forest-forming species in the calculations. It is taken into account using the procedure of discounting the annual economic effect of protective forest plantations that have reached the planned height, estimated at current prices.

Stage 5. Benefit–cost ratio (B/C).

The benefit–cost ratio [11,40] is the most capacious concept that characterizes the cost effectiveness of measures for forest reclamation of farmland. It is based on real-world estimates as an object of investment in agricultural nature management.

The B/C indicator compares discounted benefits (B) and discounted costs (C):

$$\frac{B}{C} = \frac{\sum\limits_{t=0}^{n} \frac{B_t}{(1+r)^t}}{\sum\limits_{t=0}^{n} \frac{C_t}{(1+r)^t}} \quad (1)$$

In the formula, t—benefits or costs in t year, years; n—time horizon, years; r—discount rate.

It is believed that if this ratio is greater than 1, then the cultivation of planted forests in the region is a profitable activity, respectively, less than 1 is unprofitable [12].

2.4.2. Evaluation of the Effectiveness of the Impact of Forest Plantations on Reducing Degradation and Environmental Pollution of the Adjacent agricultural Territory

The formula for assessing the ecological and economic efficiency of agroforestry has the following form:

$$E_{\frac{B}{C}} = \frac{\sum_1^T Q_t \frac{1}{(1+r)^t}}{Q_{max} \sum\limits_1^T \frac{1}{(1+r)^t}} \sum\limits_1^T \frac{Q_{eros} + Q_{dust}}{C} \quad (2)$$

In the formula, $E_{\frac{B}{C}}$ is the cost–effectiveness coefficient for land-use afforestation measures; $Q_{eros}$ is the anti-erosion effect (damage prevented by trees from soil erosion), EUR; $Q_{dust}$ is the dust-cleaning effect (damage prevented by trees from environmental dusting), EUR; C is the cost budget of the regional afforestation strategy, EUR; $Q_{max}$ is the effect of trees reached the design height, EUR.; r—discount rate; t—effect in t year, years; T—standard service life of the forest-forming breed, years.

The anti-erosion effect is expressed by the compensatory costs of replenishing the annual losses of organic and nutrient substances of the soil with fertilizers, including the costs of their delivery:

$$Q_{eros} = W_1 * V_1 + W_2 * V_2 + W_3 * V_3 + W_4 * V_4 \quad (3)$$

In the formula, $W_1$, $W_2$, $W_3$, $W_4$—the weight of humus, nitrogen, phosphorus, and potassium, respectively, contained in 1 ton of fertile soil mass translated into active substance, tons; $V_1$, $V_2$, $V_3$, $V_4$—the value of organic, nitrogen, phosphorus, and potassium fertilizers, respectively, taking into account the purchase, transportation, storage, and deposit, in EUR.

The dust-cleaning effect of forest plantations was determined using the following formula:

$$Q_{dust} = q_{dust} * P_{ment} \quad (4)$$

In the formula, qdust—the amount of dust filtered by forest plantations per year, tons; Pment—the rate of payment per ton of suspended particle emissions RM 2.5, EUR.

The cost budget for afforestation of farmland (C, EUR) includes the costs of creating protective forest stands ($C_1$), the costs of growing plantings before the closing of the crowns ($C_2$), the costs of creating the simplest hydraulic structures in the lower aisle of the forest plantation ($C_3$):

$$C = C_1 + C_2 + C_3 \quad (5)$$

The regional afforestation strategy is considered cost-effective if:

$$E_{\frac{B}{C}} \geq 1 \quad (6)$$

## 3. Results

### 3.1. Assessment of the Capital Intensity of Forest Reclamation Activities on the Slopes

By modeling bioengineered regional parameters of agroforestry landscapes within the accepted conditions, it was determined that in the territory of land use with uneven relief due to narrowing of distances between trees with an increase in slope steepness (Figure 3), protective forest cover increases almost 2 times—from 1.8 to 3.3 %. This narrowing is explained by the need to contain erosion-hazardous runoff, the intensity of which increases with distance from the watershed downhill.

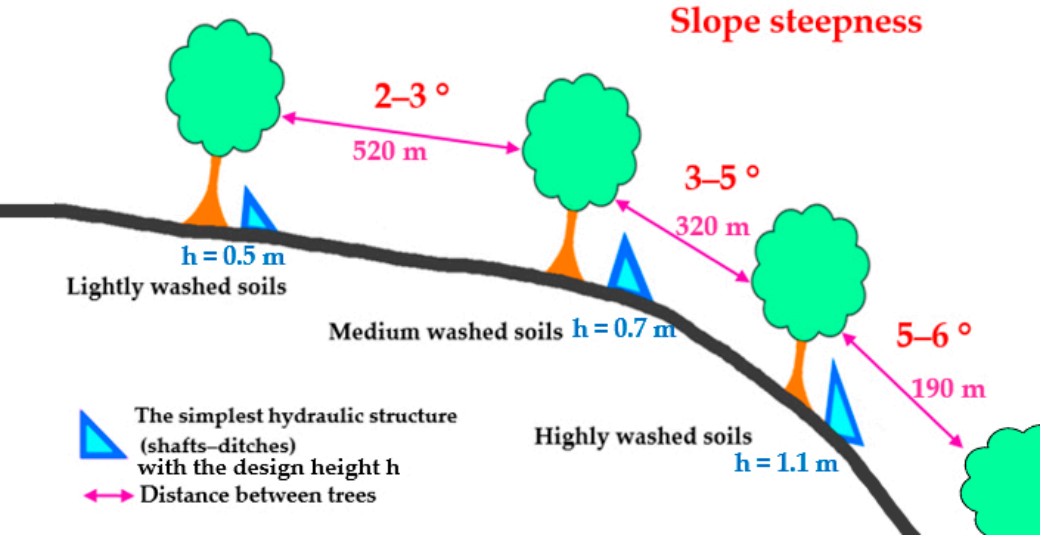

**Figure 3.** Erosion control of the slope with protective forest plantations and hydraulic structures.

Cost budgeting has established that the capital intensity of protective measures to regulate harmful runoff on slopes includes:

1. The costs of measures for the creation of forest plantations.

These measures on slopes with a steepness of 2.1–5.0° (Figure 4) consists of soil preparation according to the system of annual black steam with planter plowing (11–12% of the total cost), mechanized planting of 1-year-old seedlings, including planting material (42–45%), addition for the second year of planting (26–27%), agrotechnical soil care in rows and aisles (18–19%).

On steep slopes (5.1–6.0°), preliminary soil preparation is not carried out. At the same time, high technological efficiency of operations raises the value of planting and fertilizing seedlings and agrotechnical care by 1.2–1.8 times, compared with a flatter terrain.

2. The costs of forestry measures for the care of forest plantations of the 1st age period (before the crowns of trees are closed), including clearing and clearing, carried out for the purpose of caring for young plants.

3. Costs for the construction of the simplest hydraulic structures, including the creation of shafts or ditches of various heights in the lower aisle of the forest planting.

Calculations indicate that the total cost of the complex of works on forest-reclamation erosion control of the slope area (Table 2) depending on the steepness of the slope is EUR 2414–EUR 6302 per typical agricultural farm (on average 100 hectares) and EUR 24–EUR 63 per hectare of land equipped with protective plantations.

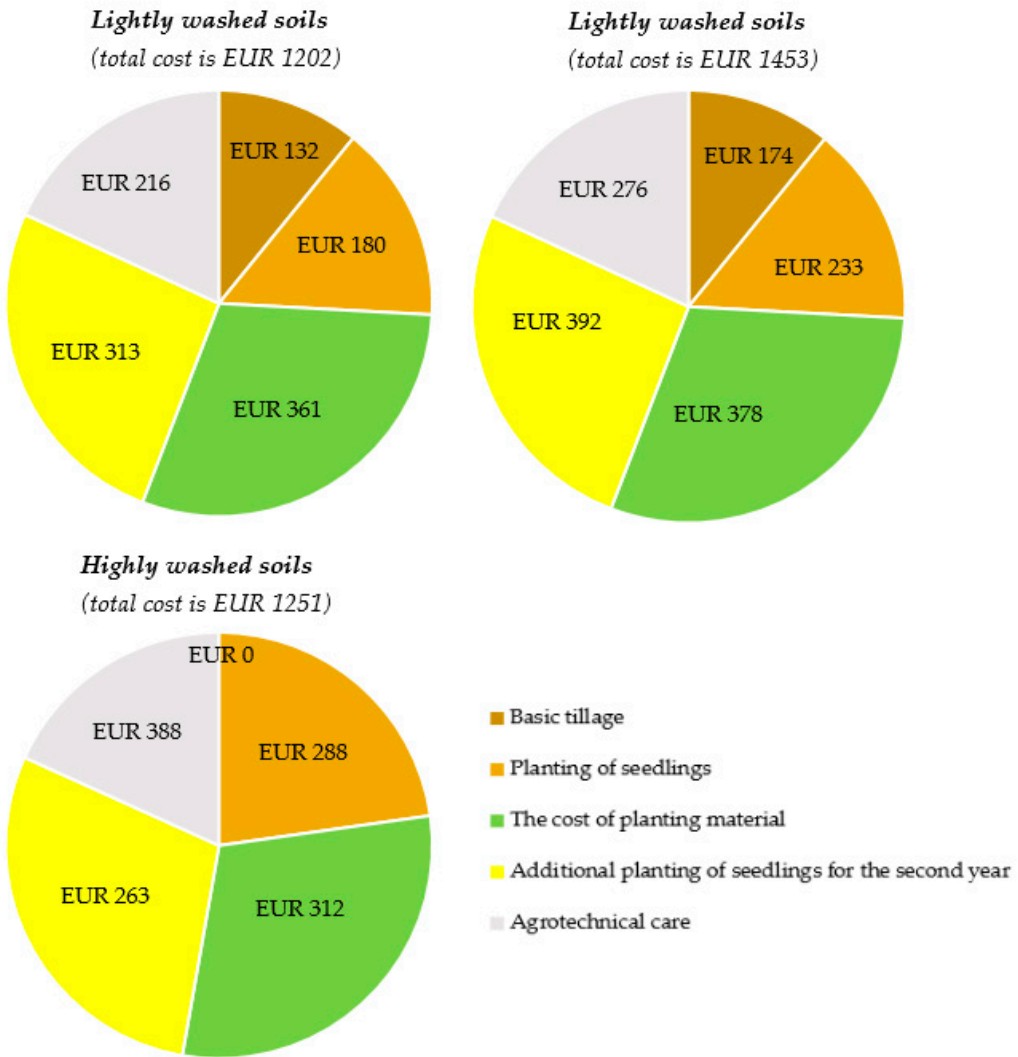

**Figure 4.** The costs of creating 1 hectare of runoff-regulating forest plantations in arid conditions of the Lower Volga region with a slope steepness of 2.1–6.0°.

**Table 2.** Capital intensity of the arrangement of the sloping agricultural landscape with forest plantations and the simplest hydraulic devices.

| Calculated Indicators | Slope Steepness, ° | | |
|---|---|---|---|
| | 2.1–3.0 | 3.1–5.0 | 5.1–6.0 |
| Land use (100 ha): | | | |
| Protective forest cover, % | 1.76 | 2.89 | 3.26 |
| The costs of creating forest plantations | EUR 2080 | EUR 4084 | EUR 3953 |
| The costs of growing plantations (logging care) | EUR 271 | EUR 321 | EUR 470 |
| The costs of creating the simplest hydraulic structures | EUR 63 | EUR 1599 | EUR 1879 |
| Total costs | EUR 2414 | EUR 6004 | EUR 6302 |
| Land use (1 ha): | | | |
| The costs of creating forest plantations | EUR 20.8 | EUR 40.8 | EUR 39.5 |
| The costs of growing plantations (logging care) | EUR 2.7 | EUR 3.2 | EUR 4.7 |
| The costs of creating the simplest hydraulic structures | EUR 0.6 | EUR 16.0 | EUR 18.8 |
| Total costs | EUR 24.1 | EUR 60.0 | EUR 63.0 |

The costs of creating forest crops have diverse dynamics—with a slope steepness from 2.1 to 5.0°, they increase by 2 times, then decrease slightly, which is explained by the peculiarities of creating forest plantations on steep slopes (5.1–6.0°). So, reducing the width of forest stands reduces the specific amount of planting material per hectare of landscape and, consequently, its cost by about 15–21%.

The costs of growing forest plantations and creating hydraulic devices increase in proportion to the increase in slope steepness ($r^2 = 0.99$)—by 1.7 times and 29.8 times, respectively. This is related to the complication of technological operations in this direction and, as a result, the rise in the cost of the 1st car-hour of vehicles on steep slopes.

In general, the costs of erosion control of land use increase with an increase in protective forest cover by 2.6 times ($r^2 = 0.98$).

*3.2. Economic Assessment of the Cost Effectiveness of Measures for the Erosion Control of Slope Lands*

Ecosystem services coming from agroforestry spread over large areas of the agricultural region and have a complex structure and determinism by natural and economic factors, but are amenable to quantitative assessment. In our opinion, on a sloping terrain, a narrow forest strip (6–9 m wide) can give the most significant positive effect in the form of proper protection of the soil from water erosion and assimilation of air pollution. The benefits of these ecosystem services are the preservation of soil fertility and improvement of public health, and their recipients are the agriculture of the region and local communities, respectively.

It is established that the economic damage that farmers will suffer during the agrotechnical development of chestnut soils on slopes without forest reclamation protection will amount to an average of EUR 637 per $ha^{-1}$ per $year^{-1}$ (Table 3). Basically, the damage is caused by the rate of degradation of these soils to the limits known to science.

**Table 3.** Ecological and economic effect obtained due to the forest management of land use on sloping plots (per ha of forested land use).

| Calculated Indicators | Open Slope without Forest Reclamation Protection | Forest-Reclamation Arrangement of Land Use at the Steepness of the Slope | | |
| --- | --- | --- | --- | --- |
| | | 2.1–3.0° | 3.1–5.0° | 5.1–6.0° |
| Erosion Hazard | | Low-Intensity Erosion | Medium-Intensity Erosion | High-Intensity Erosion |
| Annual effect of forest plantings that have reached the design height: Damage (−)/prevented damage (+) from loss of soil nutrients due to erosion Damage (−)/prevented damage (+) from air pollution by dust Annual cumulative damage/effect | EUR −584 * EUR −53 EUR −637 | EUR 284 EUR 49 EUR 333 | EUR 551 EUR 51 EUR 603 | EUR 884 EUR 56 EUR 940 |
| Average annual effect over the life of plantings when using Lanceolate ash (taking into account the time factor): Damage (−)/prevented damage (+) from loss of soil nutrients due to erosion Damage (−)/prevented damage (+) from air pollution by dust Average annual damage/effect | EUR −584 EUR −53 EUR −637 | EUR 213 EUR 37 EUR 250 | EUR 414 EUR 39 EUR 453 | EUR 663 EUR 41 EUR 704 |

Note: *—median damage on slopes with a steepness of 2.1–6.0°.

With forest reclamation protection of slope lands within the accepted conditions, the cumulative (ecological and economic) annual effect of these measures is EUR 333–EUR

940 per hectare of a field equipped with forest. This effect is a function of orographic conditions—with a raising of the steepness of the slope, along with a growth in protective forest cover, it grows rapidly (almost 3 times) ($r^2 = 0.98$).

This pattern is provided by the main anti-erosion effect (expressed in the prevented damage from soil erosion), which makes the greatest contribution (more than 90%) to the structure of the total efficiency of forest reclamation and is determined by the high cost of fertilizers needed to restore the fertility of eroded lands, especially on steep (more than 5°) slopes.

Ecosystem services of forest plantations to clean up the environment from pollution also make a certain contribution to the structure of the positive effect of forest reclamation activities in this area and amount to a considerable amount in monetary terms—EUR 49–EUR 56 per year per hectare of field protected by forest plantations in year.

The discounted value of the annual economic effect of spreading the influence of trees on the adjacent territory (taking into account the growth factor) depends on the maximum operational life of trees. So, for Lanceolate ash—the main forest-forming species used for the forest-reclamation of slopes in the dry steppe (the standard service life is 35 years), and the average annual effect is one third less than its annual value.

The effectiveness of forest reclamation measures is determined by B/C—an indicator that compares discounted benefits with discounted costs. Calculations show (Table 4) that this ratio (r = 2%) in the entire range of orographic conditions of the dry steppe of the Lower Volga region remains significantly higher than the normative value established for agriculture in Russia (0.10) and its world estimates (1.00). This convincingly proves the high reliability and expediency of capital investments in the construction of protective forest plantations on agricultural land in conditions of uneven terrain. Due to the significant damage prevented, forest reclamation is especially effective on steep slopes—B/C is equal to 11.

**Table 4.** Cost-effectiveness of protective afforestation measures on the slope lands of the dry-steppe zone of the Lower Volga region (per 1 ha of forested land use).

| Calculated Indicators | Open Slope without Forest Reclamation Protection | Forest-Reclamation Arrangement of Land Use at the Steepness of the Slope | | |
|---|---|---|---|---|
| | | 2.1–3.0° | 3.1–5.0° | 5.1–6.0° |
| Erosion Hazard | | Low-Intensity Erosion | Medium-Intensity Erosion | High-Intensity Erosion |
| Discounted costs for forest reclamation and hydraulic structures | x | EUR 23.6 | 58.8 | 61.8 |
| Discounted damage/prevented damage from chestnut soil erosion | EUR −637 | EUR 250 | EUR 453 | EUR 704 |
| Cost efficiency (benefit–cost ratio) | x | 10.6 | 7.7 | 11.4 |

It should be noted that in open areas of watersheds, the activities of agricultural farms cannot be considered economically efficient if there is a high risk of manifestation and formation of water erosion, whatever the projected level of profitability of this agricultural production.

## 4. Discussion

Soil degradation is a very large problem in modern environmental management. It is believed [41] that erosion and soil degradation have a catastrophic impact on agricultural productivity. Soil degradation is also associated with the overall quality of the environment, of which air pollution is also a major problem of global importance [42].

In many agricultural regions, soil losses exceed natural soil formation [43]. Such rates of environmental degradation require sustainable soil management methods based on an

effective nutrient cycle and reduction of degradation of natural resources while increasing the productivity of agricultural systems [44].

Agroforestry has proven itself well as a strategy for increasing and maintaining long-term productivity and soil stability [45,46]. Agroforest reclamation systems are the defining elements of rural areas and are considered as part of the working landscape. They provide ecosystem services, environmental benefits and economic goods [36,47]. Trees as such and the agroforestry system of which they are a part provide direct benefits to farmers, often through a combination of goods for the vital needs of farms, buffering climate variability and protecting soil and water resources from degradation [48]. At the same time, it is noted [49] that the value of ecosystem services provided by forests to agriculture is ignored in the literature.

While the efficient production of food and fuel is the main motivation for the creation of any agroforestry system, the benefits of soil conservation should still be recognized as their important advantage. The main justification for the creation of forest plantations in modern conditions should be the restoration of the productivity of degraded lands, increasing the productivity of low-use lands and ensuring the sustainability of agricultural environmental management. However, the adoption of an agroforestry strategy should be determined, first of all, also by economic returns.

Cost-benefit analysis is widely used to assess the profitability of agroforestry farming methods in different countries [50].

According to the data available in the literature, the costs of preparing the territory and planting forest strips amount to USD 699 ha$^{-1}$, while the cost of clearing the forest protected area ranges from USD 500 to USD 1356 ha$^{-1}$, and the cost of pruning trees is USD 148 ha$^{-1}$. [13]. According to other data, the cost of preparing the site in a mechanized way is USD 36, the cost of planting seedlings is USD 562 [51]. There are also data on the costs associated with the introduction of agroforestry per hectare, and the costs associated with the adoption of agroforestry per hectare is USD 31.4 [52].

Our data on the costs of planting a hectare of forest plantations (only the cost of planting seedlings) are consistent with those presented in the literature—they range from EUR 180 to EUR 287 per hectare and are explained by regional pricing norms in the forestry industry.

With regard to the benefits of agroforestry, when justifying environmental projects in traditional economic terms, the recognition of a significant decrease in crop yields caused by nutrient losses as a result of erosion is supported [53,54]. Thus, the direct benefits of agroforestry are estimated by prevented nutrient losses as a result of erosion, the monetary value of which is estimated as savings in fertilizer costs that would otherwise be incurred to compensate for nutrient losses as a result of soil erosion [55].

The minimum existing estimates of the cost of combating soil erosion using agroforestry are USD 1.2 per ton of soil erosion [56], as well as USD 20 per ton (methodologies based on the market price of soil for its direct use) [57].

The maximum estimates of ecosystem services to reduce soil erosion presented in the literature are USD 9.5 million year$^{-1}$ [58].

Based on our calculations, the median anti−erosion effect obtained from forest plantations as a result of preventing the loss of organic and nutrient substances of chestnut soils is EUR 438 ha$^{-1}$ year$^{-1}$ per hectare of land. The magnitude of the regional effect on dust assimilation is less—EUR 53 ha$^{-1}$ year$^{-1}$.

Our estimates of the benefit-cost ratio (9.4–13.9) are within the range of values available in the literature. So, in the study [59], the profitability of agroforestry technologies in semi-arid areas of the Western Pokot district, calculated on the basis of B/C, ranges from 0.68 to 9.40. The ratio depends on the technology of agroforestry adopted by the farmer and the main types of economic activity at the location of the farmer.

In other studies, all agroforestry systems were more profitable than swidden systems. Their benefit–cost ratios were calculated in the amount of 10.36–16.19 [12].

Taking into account the environmental effects of agroforestry increases the ratio of B/C by 3–6 times, compared with its commercial effects (net income from the sale of agricultural and wood products) [60].

## 5. Conclusions

The study was conducted to assess the effectiveness and expediency of measures to create protective forest systems on sloping lands in arid conditions. The calculated data on the effectiveness of agroforestry systems, the costs of planting trees, their cultivation, as well as the installation of a hydraulic element in the aisle, the potential return on these investments in the form of damage prevented by forest plantations from loss of soil nutrients and from air pollution by dust are very valuable in the study.

Based on the modeling of various tree placement options in an agricultural landscape with uneven terrain, it was found that on sloping lands, the value of the annual economic effect from ecosystem services of protective forest stands strictly depends on the slope steepness and the amount of soil flushing that trees prevent. At the same time, the discounted value of this effect is one third less than its annual value. It is proved that with the optimal placement of trees on sloping lands (when the distances between them decrease when moving away from the watershed downhill), the benefit–cost ratio for each option exceeds 1.

In order to make financial sense of regional measures to create protective forest plantations, it is necessary to develop appropriate markets that would offer compensation for ecosystem services provided by trees. The information provided can serve as a guide in the procedure of quoting the costs of agroforestry.

It is possible to practically implement the results of the study by implementing given forest-reclamation strategies in farm systems with degraded soil cover. The main motivating indicator of the creation of forest plantations for land users in this case may be the opportunity presented in the article to see the benefits in advance in the form of an even annual value of the damage prevented from erosion and air pollution. A clear demonstration of the ability of agroforestry systems to improve the condition of soils and the environment in the form of long-term economic benefits can influence the decision of decision makers to subsidize farmers. By supporting farmers, the state will support the sustainable development of the country, increasing—through agroforestry—the level of environmental sustainability of agricultural land subject to degradation.

**Author Contributions:** Conceptualization, E.A.K.; methodology, E.A.K.; software, E.A.K.; formal analysis, E.A.K.; resources, A.I.B.; project administration, A.I.B. All authors have read and agreed to the published version of the manuscript.

**Funding:** The article has been prepared in accordance with the state task of the Russian Ministry of Education and Science No. FNFE-2022-0015 to Federal Scientific Center of Agro-ecology, Complex Melioration and Protective Afforestation Russian Academy of Sciences.

**Institutional Review Board Statement:** Not applicable.

**Informed Consent Statement:** Not applicable.

**Data Availability Statement:** Data available on request.

**Conflicts of Interest:** The authors declare no conflict of interest.

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
