# Peer review of "Assessment of Ecological and Economic Efficiency of Agroforestry Systems in Arid Conditions of the Lower Volga"

_forests, doi:10.3390/f13081248_

Round 1

Reviewer 1 Report

To Forests

Dear Editor,

Thanks for allowing me to review the manuscript titled 'Assessment of Ecological and Economic Efficiency of Agroforestry Systems in Arid Conditions of the Lower Volga’ submitted to forests.

I have the following major and minor comments for the Author of the manuscript.

General comment

-          The abstract part is too long and focuses on the major findings. The points you described starting to line,,,,,,, the aim of this study …. onwards is enough and the other sections above this points are not part of your study results and it is a description of the general knowledge

Materials and methods section,

-          In the case study site (section 2.1), the area is not well described. I recommend you include a map of the study area and include also climatic graphs (temperature and rainfall graphs) of the study area. In this section, the sentence the sum of active temperatures is 2500 – 3100 C is not clear. Temperature data should be described as maximum and minimum temperature and describe clearly.

-          The agroforestry practice used in the area is not well described. It is mostly described as an afforested area by Lanceolate ash (Fraxinus lanceolate). If it is an agroforestry system what type is it, is it wind break? agro-silvopasture? what type of crops are mixed with the tree species?. Clearly describe it

-          The method of data collection part is not clear. I did not see a replication for the data collection to evaluate the losses of humus and nutrients, anti-erosion effects, dust filtered by forest stands in the agroforestry system, etc. All these have to be described clearly. If you used secondary data, it has also to be clear. Overall, I did not see a replication. Even if the author indicated there is an agroforestry system, there is a variation in their effects in anti-erosion effects, ecosystem services, etc. within the studied system

Results

-          The results part has to be improved. Focus to present only your findings. It seems the result and discussion part are mixed. In the results section, there is no need to put references. Describe only your results.

-          Under section 3.1, paragraph 1, 2, 3, and 4 seems a description of the study area. It is not a result

Discussion

-          In this part, you presented more on the general knowledge of agroforestry's role in ecosystem services, productivity etc. This section requires lots of improvements. Discuss your findings in a respective of other similar study results.

-          I recommended you to re-write the discussion part.

Conclusion

-          In this section describe on what are your major findings. This section is too long and some of the paragraphs you described are not related to your findings, it is a piece of general knowledge. Re-write and improve it.

Specific comments

-          Improve some of the grammatical errors such as

1.   In the abstract part, you used B/C. Before you use abbreviations write the full name that is the benefit-cost ratio (B/C)

2.  The word per 1 hectare is repeatedly described in different sections of the paper and it is wrong. Correct it by writing as per hectare

3.       Write r2=98%  as r2=0.98 

Reviewer 2 Report

The manuscript is interesting and the subject of the research is quite innovative. However, there are some points, mainly methodological, that should be improved. In many places it is not clear how the findings are obtained.

 Firstly, the author mention that she restricted her research to “the regulatory ecosystem services of trees - control of erosion and atmospheric pollution in the specific context of agricultural systems.” However, from this derive a limitation – underestimation of the value of the ecosystem services. It would be beneficial for the research to taken into account additional services.

Secondly, the author mention “The cost equivalent of these ecosystem services was estimated using the known methods [31, 32] at prices of 2022”, but it is not at all clear what the methods are. The 3 references cited describe different methodologies. It is also necessary for the reader to have at least a brief description of the methodology applied.

Similarly, the manuscript includes a general definition of “Benefit-сost ratio”, but there is no description of the variables which taken into account in the implementation of the method. That is the core of a “Benefit-сost ratio” implementation.

There are numerous typographical errors (double dots, unnecessary spaces, etc.).

Please correct “2.1.С. ase Study Sites”

Section Materials and Methods, line 10: Please add a full stop.

Reviewer 3 Report

The article presents an interesting study on the benefits of agroforestry and effectiveness of measures to create protective forest systems. What remains unclear is how the findings can be practically implemented to incentivise land owners and thus create motivation to achieve the expected sustainability results ( in particular as not all land owners might be equally effected or equally able to contribute their share without seeing an immediate gain).

Round 2

Reviewer 1 Report

Thanks. I have gone through the revised manuscript titled 'assessment of Ecological and Economic Efficiency of Agroforestry'. All the comments I provided in are included in the manuscript and it is improved. Therefore, I recommend to accept the manuscript. Regards,

Reviewer 2 Report

Reading carefully the corrections made by the authors, I find that there is no significant improvement of the text. Discussion and conclusion sections are very similar to the original text. The novelty of the present work has not been identified, as I suggested. Generally, I think that the manuscript was improved rather superficially.

My main concern remains, that the manuscript is quite similar to the paper titled “Economic Assessment and Management of Agroforestry Productivity from the Perspective of Sustainable Land Use in the South of the Russian Plain”. And this fact weakens the originality of the submitted manuscript.
